# Enhanced prediction of thrombotic events in hospitalized COVID-19 patients with soluble thrombomodulin

Sergio Padilla[1,2☉]*, María Andreo[3☉], Pascual Marco[4], Ana Marco-Rico[4], Christian Ledesma[5], Marta Fernández-González[2,5] Javier García-Abellán[1,2], Paula Mascarell[1,2], Ángela Botella[5], Félix Gutiérrez[1,2¶], Mar Masiá[1,2¶]*

1 Infectious Diseases Unit and Department of Clinical Medicine, Hospital General Universitario and Universidad Miguel Hernández de Elche, Alicante, Spain, 2 CIBER de Enfermedades Infecciosas (CIBERINFEC; Biomedical Research Networking Center for Infectious Diseases), Instituto de Salud Carlos III, Madrid, Spain, 3 Internal Medicine Service and Department of Clinical Medicine, Hospital General Universitario and Universidad Miguel Hernández de Elche, Alicante, Spain, 4 Hematology Department, Hospital General Universitario Dr. Balmis, Alicante, Spain, 5 Infectious Diseases Unit, Hospital General Universitario de Elche, Alicante, Spain

¶Félix Gutiérrez and Mar Masiá are joint senior authors and made equal contribution to the article.
☉ These authors contributed equally to this work.
* spadilla@umh.es (SP); marmasiac@gmail.com (MM)

## Abstract

We aimed to determine the predictive accuracy of elevated soluble thrombomodulin (sTM) and angiopoietin-2 (Ang2) for thrombotic events (TE) in hospitalized COVID-19 patients. We conducted a nested case-control study within a cohort of people admitted to hospital with COVID-19 from March 2020 to August 2022. The cases (people with TE within 28 days after hospital admission) were matched by propensity score to comparable patients without TE. We determined plasma levels of sTM and Ang2 in all available frozen samples, prioritizing the earliest post-admission samples, using an automated immunoassay technique. Among 2,524 hospitalized COVID-19 patients (43% females; median age 67 years), 73 had TE (incidence 1.15 events per 1000 patient-days of follow-up). Frozen plasma samples were available for 43 cases and 176 controls. Elevated plasma concentration of sTM was significantly associated with TE (2.8 [1.8, 4] vs. 1.52 [1.1, 2.65] ng/mL; p = 0.001) and mortality (median [Q1, Q3], 3.32 [2.16, 4.65] vs. 1.58 [1.11, 2.73] ng/mL; p = 0.001), while D-dimer showed a specific association with TE (2.3 [0.8, 7.4] vs. 0.75 [0.4, 1.6] mcg/mL; p = 0.001). In contrast, Ang2 was not associated with any of these events. The association with thrombotic events remained in adjusted models (HR [95%CI] per unit increase, 1.24 [1.04-1.47] for sTM; 1.07 [1.03-1.10] for D-dimer). The adjusted regression model that included both biomarkers, sTM and D-dimer, improved (AUC 73%, sensitivity 77% and specificity 65% for TE diagnosis; p = 0.007) the predictive capacity of the same model without sTM. In conclusion, determination of soluble thrombomodulin along with D-dimer enhances thrombotic risk assessment in hospitalized COVID-19 patients.

**Data availability statement:** The datasets generated and/or analysed during the current study are available in the Mendeley Data public repository, https://data.mendeley.com/datasets/dr65hp957x/1

**Funding:** This work was supported by Spanish National Plan for Scientific and Technical Research and Innovation, European Regional Development Fund (ERDF) and Instituto de Salud Carlos III (RD16/0025/0038, PI16/01740, PI18/01861, CM19/00160, CM20/00066, COV20/00005, CM21/00186, CM22/00026 and PI22/01949); Consorcio Centro de Investigación Biomédica en Red (CIBER), Instituto de Salud Carlos III, Ministerio de Ciencia e Innovación and European Union – NextGenerationEU (CB21/13/00011); ILISABIO programme, UMH-FISABIO, Generalitat Valenciana (A-32 2020); Consellería de Innovación, Universidades, Ciencia y Sociedad Digital, Generalitat Valenciana (AICO/2021/205); and funded by Conselleria de Sanitat Universal i Salut Pública (Generalitat Valenciana, Spain) and the EU Operational Program of the European Regional Development Fund (ERDF) for the Valencian Community 2014–2020, within the framework of the REACT-EU program, as the Union's response to the COVID-19 pandemic (FEDER-COVID-23). The funders had no role in study design, data collection and analysis, decision to publish, or preparation of the manuscript.

**Competing interests:** The authors have declared that no competing interests exist.

## Introduction

Early in the COVID-19 pandemic, it became evident that cardiovascular events, particularly thrombotic events, were closely associated with morbidity and mortality in people infected with SARS-CoV-2. Relevant events included both arterial thrombosis, such as myocardial ischemia [1], arteritis [2] and venous thromboembolism (VTE), with reported incidences of VTE ranging from 17-27% in patients admitted to intensive care units (ICU) [3–5], and occurring significantly in general hospital wards as well [6]. Thrombotic events are a diagnostic priority owing to their severity and potential preventability [1–4,6,7].

Unlike typical thrombosis, often linked to traditional risk factors, COVID-19-related thrombosis frequently occurs in normally healthy individuals. The prevailing theory, as anticipated in studies on SARS-CoV-1 infection [8], suggests that SARS-CoV-2 invades vascular endothelial cells, causing vascular damage and initiating a systemic immune response, which leads to immunothrombosis [9–11]. This hypercoagulable state, associated with high mortality rates [12], is mediated by inflammatory phenomena, endothelial dysfunction, and platelet activation [13,14].

Multiple clinical trials across various settings, pharmacological strategies, doses, and clinical endpoints have demonstrated the benefit of anticoagulation and/or antithrombotic prophylaxis in mitigating venous and arterial events [15,16]. Notably, some studies have incorporated interventions based on elevated levels of hypercoagulability biomarkers such as D-dimer [16,17]. However, D-dimer testing still presents serious issues related to standardization, harmonization, and precise knowledge of assay performance characteristics [18]. Moreover, D-dimer testing has limitations in cases of functional fibrinolytic deficiency: in people with severe COVID-19, decreased fibrin clot lysis and increased resistance to fibrinolysis are inaccurate measures of clotting extent [19]. Lastly, there is no evidence on the accuracy of D-dimer determination for predicting arterial thrombosis [20], a significant and unpredictable condition in COVID-19 patients that lacks effective prophylactic strategies. Healthcare providers must take these limitations into account when using D-dimer levels to detect thrombotic events in people with COVID-19.

Although soluble thrombomodulin (sTM) can be elevated in various other processes involving endothelial injury, research suggests that one important disease mechanism in COVID-19 is transition to a procoagulant state of endothelial cells characterized by elevated levels of circulating soluble thrombomodulin (sTM) [21]. sTM is a cleaved product of full-length thrombomodulin (TM), released from the endothelial surface during inflammation [22]. In parallel, elevated plasma angiopoietin-2 (Ang2), an inflammatory cytokine, is a strong predictor of death in infection-mediated acute respiratory distress syndrome and correlates with COVID-19 disease severity [23,24]. Importantly, novel in vivo evidence in COVID-19 patients suggests that Ang2 interacts directly with the coagulation system by binding and inhibiting TM-mediated anticoagulation [23]. This highlights a potential shared mechanism involving both Ang2 and TM as early biomarkers for thrombosis in COVID-19.

The aim of this study was to identify enhanced and novel predictors of thrombotic events in a large cohort of people hospitalized with COVID-19, improving upon the predictive capabilities of D-dimer.

## Methods

We conducted a prospective longitudinal study in Hospital General Universitario de Elche (Alicante, Spain), including all COVID-19 cases confirmed by real-time polymerase chain reaction from nasopharyngeal swab samples, among people admitted between 1 March 2020 and 31 July 2022. Data for research purposes were accessed on 19/10/2023.

During the COVID-19 pandemic, hospitalized COVID-19 patients were managed in accordance with a local protocol that outlined specific diagnostic and therapeutic procedures [25], including active surveillance of thrombotic events. In individuals with poor initial response (S1 Figure in S1 File) and a more than twofold increase in D-dimer levels, the protocol recommended computed tomographic pulmonary angiography (CTPA) to rule out pulmonary artery thrombosis (PAT). Four-extremity doppler ultrasound was not routinely performed but was used at the physician's discretion for suspected deep venous thrombosis (DVT). If CTPA was not feasible, the patient was started on empirical anticoagulation until the diagnosis could be confirmed or excluded.

From 21 April 2020, the Thrombosis and Hemostasis Commission of our hospital recommended thromboprophylaxis for people admitted with COVID-19 (subcutaneous enoxaparin at a dose of 40 mg/24h, or 60 mg/24h for people weighing over 80 kg). In cases of high thrombotic risk, the enoxaparin dose was increased to 1 mg/kg/day. The criteria for high thrombosis risk included: D-dimer levels above 1.5 µg/ml, plasma interleukin-6 levels above 40 pg/mL, lymphocytes below 800 x10⁹/L, and/or serum ferritin levels above 1000 µg/L. Contraindications for the use of low molecular weight heparin (LMWH) included significant bleeding and severe thrombocytopenia (< 25,000 platelets/µL). Patients already using anticoagulants prior to admission were to continue therapy, preferably switching to the corresponding dose of enoxaparin or intravenous heparin. There were no recommendations regarding the initiation or discontinuation of antiplatelet therapy.

The protocol also included standardized collection of clinical data and serial collection of blood samples at hospital admission and at various time points during hospitalization for biochemical and sero-virological measurements.

## Ethics

The ethics committee of our hospital approved this study as part of the COVID-19 Elx/Spain project. Written informed consent for participation was unnecessary, according to Spanish legislation and institutional requirements (Ethical Committee of the Hospital General Universitario de Elche (Spain)). All study procedures were conducted in accordance with the Good Clinical Practice guidelines, the principles of the Declaration of Helsinki, and local laws.

## Plasma coagulation biomarkers determination

Plasma levels of sTM and Ang2 were measured in frozen samples at the Coagulation Laboratory of the Dr. Balmis General University Hospital in Alicante, Spain. Blood was collected in K2-EDTA tubes and centrifuged. Plasma was then aliquoted and stored at −80 °C. For individuals in the propensity score nested cohort, determination of sTM and Ang2 levels was conducted in two batches of thawed samples at 37ºC in a water bath. sTM (Abcam, Cambridge, UK, ELISA kit, Cat #ab46508; range from 0.625-20 ng/mL) and Ang2 (Abcam, Cambridge, UK, Angiopoietin 2 Human ELISA kit, Cat #ab99971; range from 4.12-3000 pg/mL) plasma levels were measured by Enzyme-Linked Immunosorbent Assay carried-out in an automated ELISA device (Dynex DS2® ELISA system), following protocols from the manufacturer.

Plasma D-dimer testing was carried out on fresh samples at the local Clinical Laboratory of the General University Hospital of Elche, Spain. The same methods, reagents and instruments were used throughout the study period (automated immune-turbidometric assay using INNOVANCE® D-dimer kits [Siemens Healthinners] on Sysmex CS-2500 System [Siemens]).

In patients with thrombotic events, the sample closest to the event, prioritizing those collected prior to it, and in the controls, the earliest sample obtained after hospital admission, were selected for biomarker determination.

## Statistical analyses

We used descriptive statistics to summarize the baseline characteristics of the cohort: median with first and third quartile for continuous data, and absolute and relative frequencies for categorical data. We used the Wilcoxon test to compare groups in continuous variables and the χ2 test or Fisher's exact to compare categorical groups. To calculate incidence rates (IRs) we divided the number of observed events by the total person-time at risk, expressed per 1,000 person-days. We used the Wald method to calculate the corresponding 95% confidence intervals (CIs), accounting for the observed number of events and person-time.

To determine the predictive value of sTM, Ang2 and D-dimer for thrombotic events, we selected all patients with thrombotic events and used propensity scores to match them in a 1:3 ratio to a comparable group without thrombotic events. The covariates employed for propensity matching were sex, age, Charlson comorbidity index, and World Health Organization (WHO) COVID-19 severity score. We adopted this approach with the aim of achieving comparable groups and reliable analytical outcomes. To assess similarity between the groups, we examined changes in standardized mean differences (SMD) of the variables. Although follow-up extended beyond 90 days, our focus was on thrombotic events occurring within the initial 28 days after hospital admission. This was because we wanted to exclude cases of thrombosis potentially related to medical factors other than SARS-CoV-2 infection (e.g., reduced mobility or medical procedures).

The main analyses involved Cox proportional hazards regression modeling. We included potential confounders in the models, including baseline variables selected based on statistical and clinical significance. Variables used for propensity score matching (age, sex, Charlson comorbidity index, and WHO COVID-19 severity score) were not reintroduced in the models to avoid collinearity and over-adjustment. To enhance reliability and address potential limitations associated with finite sample sizes we employed bootstrap resampling when estimating model parameters and CIs. To assess the diagnostic performance of specific biomarkers (or combinations of biomarkers) for thrombotic events, we used receiver operating characteristic (ROC) curves. Statistical analyses were performed using R software (R-Core Team 2020, R-4.1.2.1). Because there were few missing values for all variables, we used case-wise deletion.

## Results

### Baseline characteristics and clinical status upon admission

Between 1 March 2020 and 31 July 2022, 2,524 patients were admitted to our hospital with SARS-CoV-2 infection, accounting for 63,560 patient-days of cumulative cohort follow-up at 28 days. The median (Q1, Q3) age at admission was 67 (54, 80) years, and 43% of the sample were females. Of all COVID-19 patients, 4% had severe COVID-19 at hospital admission (WHO COVID-19 severity score > 4), 8% required ICU admission, and 9% died (from any cause). Regarding SARS-CoV-2 variants, there were 552 (22%) Omicron infections, 1954 (78%) pre-Omicron infections (1224 Ancestral, 511 Alpha, 219 Delta), and 18 cases of unknown variants (S1 Table in S1 File).

### Characteristics and frequency of thrombotic events

During the first 28 days after hospital admission, 256 patients (10.5%) had CTPA (with a positivity rate of 22%) and 40 (1.6%) had bilateral compression ultrasound of the legs (positivity rate of 43%). These tests led to the diagnosis of 56 venous thrombotic events including 46 cases of PAT and 10 of DVT without PAT. In addition, there were 17 diagnoses of arterial thrombosis (AT; nine ischemic strokes, six acute myocardial infarctions and two cases of acute

peripheral ischemia), bringing the total to 73 thrombotic events (29 Ancestral variant, 28 Alpha, 10 Delta, and 6 Omicron). The IR [95% CI] per 1,000 person-days was 1.15 [0.90-1.44] for thrombotic events, 0.72 [0.53-0.96] for PAT, and 0.16 [0.08-0.29] of DVT without PAT. The intra-Hospital IR [95% CI] of thrombotic events was 3.25 [2.49-4.18] per 1,000 person-day among all patients, and was highest in patients with a score of 7 point on the WHO COVID-19 severity scale (4.34 [0.53-15.67]) (S2 Table in S1 File). Most cases of thrombosis were observed within the first five days following hospital admission (S2 Figure in S1 File). An additional twenty events occurred between days 29 and 90, but these were not included in the analysis.

Of the PAT identified, 24% involved central regions. The Pulmonary Embolism Severity Index (PESI) [26] indicated high risk of 28-day mortality (PESI category IV) in 28% of patients with PAT, and very high risk (PESI category V) in 17% (S3 Table in S1 File). Occurrence of venous and arterial thrombotic events (compared with occurrence of no events) was associated with increased need for noninvasive ventilatory support (26% vs. 8%; p = 0.001) and invasive ventilatory support (21% vs. 5%; p = 0.001), and ICU admission (36% vs. 7%; p = 0.001); but no significant increase in overall mortality (15% vs. 9%; p = 0.138). S1 Table in S1 File shows the occurrence of thrombotic events according to these and other baseline characteristics.

## Analysis of coagulation plasma biomarkers

We matched the 73 cases (patients with thrombotic events within 28 days of admission) to 297 paired patients without thrombotic events, effectively reducing the SMDs in the variables used for matching (S3 Figure in S1 File). Univariate baseline predictors of thrombotic events in this sample are shown in Table 1.

Frozen plasma samples were available in 43 cases (29 PAT, 6 DVT, 8 AT) and 176 matched controls. Plasma samples analyzed in controls were collected at a median [Q1, Q3] of 1 [1, 1] days after hospital admission, and in cases, at a median [Q1, Q3] of 0 [-1, 4] days before the thrombotic event.

The three coagulation parameters studied (sTM, Ang2 and D-dimer), showed a weak positive correlation, with significant associations observed between sTM and both D-dimer and Ang2, while no correlation was observed between D-dimer and Ang2 (S4 Figure in S1 File). sTM levels were above the limit of detection in all the study patients. The D-dimer levels were above the most common clinical cut-offs of 0.5 μg/mL and 1.0 μg/mL in 68.2% and 38.2% of the patients, respectively. Unlike D-dimer, sTM and Ang2 were directly associated with age and the presence of comorbidities; only sTM showed the most consistent and significant positive correlations with inflammatory parameters (IL-6, C-reactive protein and ferritin), compared to D-dimer and Ang2 (S4 Table in S1 File). There was no difference in sTM levels between omicron and non-omicron variants (median [Q1, Q3] ng/mL, 1.95 [1.21, 2.80] vs.1.61 [1.13, 2.85]) but Ang2 levels were significantly higher in patients infected with the omicron variant (median [Q1, Q3] pg/mL, 734 [530, 1056] vs.360 [117, 837]; p < 0.001).

Univariate analyses showed that elevated plasma concentration of sTM was significantly associated with mortality (median [Q1, Q3], 3.32 [2.16, 4.65] vs. 1.58 [1.11, 2.73] ng/mL; p = 0.001) and thrombotic events (2.8 [1.8, 4] vs. 1.52 [1.1, 2.65] ng/mL; p = 0.001). D-dimer showed a specific association with thrombotic events (2.3 [0.8, 7.4] vs. 0.75 [0.4, 1.6] mcg/mL; p = 0.001) (Fig 1).

In contrast, Ang2 showed a slight, non-significant trend towards lower levels in patients with thrombotic events (median [Q1, Q3], 310 [92, 590] vs. 502 [151, 964] ng/mL; p = 0.078) and no association with mortality (545 [112, 895] vs. 452 [139, 897] ng/mL; p = 0.870). The

**Table 1. Baseline predictors of 28-day thrombotic events in the propensity score cohort.**

|  | Thrombotic events | | | | |
|---|---|---|---|---|---|
|  | **All** | **No** | **Yes** | **Unadjusted-OR** | **P-value** |
| **N** | 370 | 297 | 73 | – | – |
| **Female sex** | 159 (43) | 127 (43) | 32 (44) | 1.04 (0.62-1.75) | 0.868 |
| **Age, years** | 72 (61, 79) | 73 (61, 79) | 72 (61, 79) | 1.00 (0.98-1.03) | 0.703 |
| **Charlson Comorbidity Index** | 4 (2, 5) | 4 (2, 5) | 4 (2, 5) | 1.01 (0.90-1.13) | 0.880 |
| **Any comorbidity** | 314 (85) | 254 (85) | 61 (84) | 0.86 (0.44-1.80) | 0.673 |
| **Cardiovascular disease** | 150 (41) | 122 (41) | 28 (38) | 0.89 (0.52-1.50) | 0.656 |
| **Hypertension** | 203 (55) | 167 (56) | 36 (49) | 0.75 (0.45-1.27) | 0.288 |
| **Diabetes** | 111 (30) | 95 (32) | 16 (22) | 0.59 (0.31-1.08) | 0.095 |
| **WHO COVID-19 severity score (5-6-7)** | 66 (17.8) | 54 (18) | 12 (16) | 0.88 (0.43-1.71) | 0.727 |
| **$FiO_2$ (%), log10** | 28 (28, 35) | 28 (28, 35) | 28 (28, 36) | 3.02 (0.46-17.76) | 0.229 |
| **eGFR (mL/min/1.73 m²), log10** | 87 (63.5-99.6) | 87 (63, 99) | 89.5 (64.5, 98) | 1.27 (0.42-4.49) | 0.991 |
| **C-reactive protein (mg/dL), log10** | 53 (20, 116) | 53 (21, 116) | 44 (9.9, 113) | 0.67 (0.47-0.96) | 0.027 |
| **IL-6 (pg/mL), log10** | 49 (15, 168) | 51 (14.6, 159) | 36 (19, 222) | 1.19 (0.82-1.75) | 0.363 |
| **Ferritin (ng/dL), log10** | 332 (143, 662) | 332 (144, 664) | 291 (134, 586) | 0.75 (0.44-1.26) | 0.277 |
| **LDH (U/L), log10** | 255 (207, 335) | 254 (204, 322) | 275 (229, 426) | 2.99 (0.99-8.87) | 0.049 |
| **NLR** | 5.2 (3, 9.9) | 5 (3, 9.5) | 6.1 (4, 12.4) | 1.02 (1.00-1.05) | 0.041 |
| **s-TM, ng/mL***  | 1.70 (1.14, 2.84) | 1.52 (1.1, 2.6) | 2.8 (1.8, 4.0) | 1.39 (1.14-1.69) | 0.001 |
| **Angiopoietin-2, (pg/mL), log10***  | 456 (135, 897) | 499 (154, 959) | 310 (92, 590) | 0.79 (0.53-1.21) | 0.243 |
| **D-dimer, μg/mL** | 0.82 (0.46, 1.93) | 0.75 (0.4, 1.46) | 2.3 (0.8, 7.4) | 1.12 (1.07-1.19) | 0.001 |
| **INR** | 1.1 (1.0, 1.2) | 1.1 (1, 1.2) | 1.1 (1, 1.2) | 0.81 (0.46-1.15) | 0.326 |
| **Platelets, (x 10³/μL), log10** | 182 (143, 236) | 175 (141, 226) | 214 (169, 289) | 1.19 (2.87-52.74) | 0.001 |
| **RT-PCR Cycle threshold** | 25 (20, 31) | 24 (19, 30) | 29 (22, 33) | 1.06 (1.02-1.11) | 0.004 |
| **SARS-CoV-2 vaccination** | 65 (18)) | 52 (18) | 13 (18) | 1.02 (0.50-1.94) | 0.952 |
| **Clinical events at 28 days** |  |  |  |  |  |
| Overall mortality | 41 (11) | 30 (10) | 11 (15) | 1.58 (0.72-3.24) | 0.228 |
| Mechanical ventilation | 60 (16) | 45 (15) | 15 (21) | 1.45 (0.74-2.73) | 0.264 |
| NIV or HFO | 61 (17) | 42 (14) | 19 (26) | 2.14 (1.14-3.92) | 0.016 |
| ICU admission | 77 (21) | 51 (17) | 26 (36) | 2.67 (1.50-4.68) | 0.001 |

*Soluble Thrombomodulin and Angiopoietin-2 plasma levels were determined in the subset of individuals with available samples (N = 219 [176 without events and 43 with thrombotic events]). Data are presented as percentages for categorical variables and medians with interquartile ranges for continuous variables. WHO, World Health Organization; $FiO_2$, fraction of inspired oxygen; eGFR, estimated glomerular filtration rate; IL-6, interleukin 6; LDH, lactate dehydrogenase; NLR, neutrophil-to-lymphocyte ratio; s-TM, soluble Thrombomodulin; INR, International Normalized Ratio; RT-PCR, reverse transcriptase polymerase chain reaction; NIV, non-invasive ventilation; HFO, high-flow oxygen; ICU, intensive care unit; OR, odds ratio. All values were measured at hospital admission except for D-dimer and sTM, which were determined closest to the thrombotic event in cases (prioritizing pre-event samples) and at the earliest available sample after admission in controls, as specified in the Methods section.

association of higher sTM and D-dimer levels with thrombotic events remained in adjusted models (HR [95%CI] per unit increase, 1.24 [1.04-1.47] for sTM; 1.07 [1.03-1.10] for D-dimer) (Fig 2).

In univariate models, the discriminative ability of both parameters for predicting thrombotic events was significant: sTM had an HR [95% CI] of 1.31 [1.13-1.53 (p < 0.001); sensitivity (se) = 64%, specificity (sp) = 70%] and D-dimer had an HR of 1.08 [1.05-1.11 (p < 0.001); se = 67%, sp = 74%]. At a cut-off point of 0.2, the adjusted regression model including both variables (sTM and D-dimer) showed higher specificity and significantly improved predictive capacity (AUC = 73%, p = 0.007; se = 77% and sp = 65%) compared with the same model without sTM (AUC = 66%; se = 66% and sp = 60%) (Fig 3).

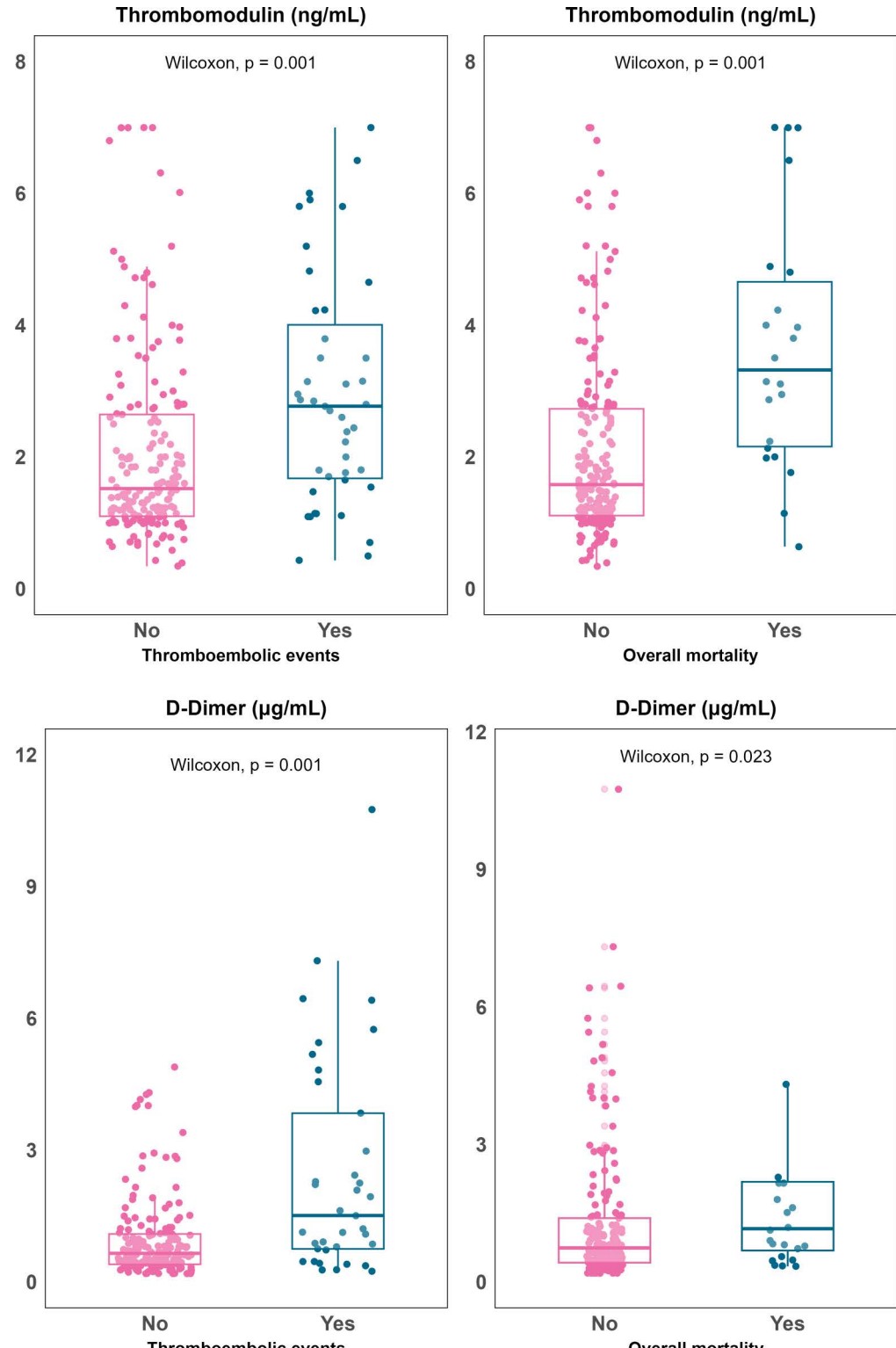

**Fig 1. Boxplots illustrating soluble thrombomodulin (Panel a) and D-dimer (Panel b) plasma concentration in relation to 28-day mortality (right) and thromboembolicevents (left).** All patients were accounted for within the 28-day follow-up period, with no losses to follow-up before the censoring date.

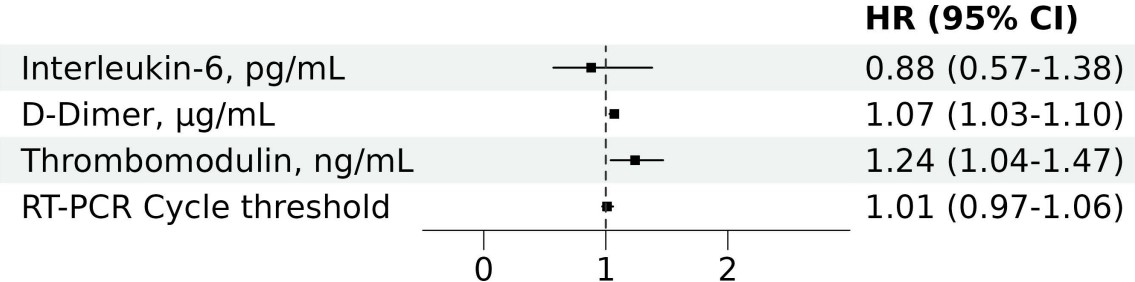

|  | | | HR (95% CI) |
|---|---|---|---|
| Interleukin-6, pg/mL | | | 0.88 (0.57-1.38) |
| D-Dimer, µg/mL | | | 1.07 (1.03-1.10) |
| Thrombomodulin, ng/mL | | | 1.24 (1.04-1.47) |
| RT-PCR Cycle threshold | | | 1.01 (0.97-1.06) |

**Fig 2. Multivariate Cox proportional hazards regression analysis of baseline predictors for 28-day thrombotic events in the propensity score cohort.** HR, Hazard Ratio; CI, confidence interval; RT-PCR, reverse transcriptase polymerase chain reaction. The figure presents adjusted HRs with 95% CIs for baseline predictors included in the multivariate model: soluble thrombomodulin (sTM), D-dimer, interleukin-6, and RT-PCR cycle threshold. The cohort was also matched by propensity score for age, sex, Charlson comorbidity index, and WHO COVID-19 severity score.

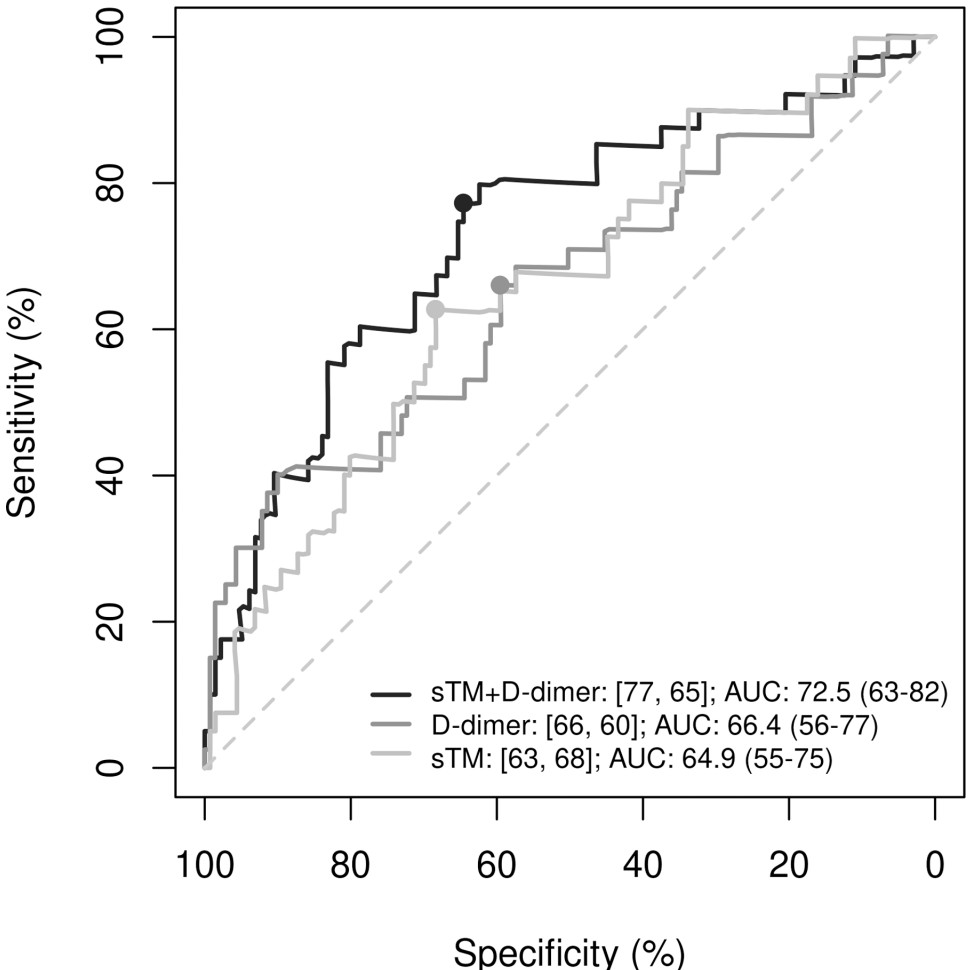

**Fig 3. Comparison of ROC curves for the prediction of thrombotic events using D-dimer, soluble trombomodulin (sTM), and their combination.** Evaluation of the predictive performance of D-dimer alone, sTM alone, and the combination of D-dimer with sTM in hospitalized COVID-19 patients. ROC, receiver operating characteristic; AUC, area under the ROC curve; s-TM, soluble thrombomodulin. In the legend, the first number represents sensitivity (%) and the second specificity (%). The cutoff value for the regression models in all three cases was 0.2. The p-value for the comparisons between the combined model (D-dimer + sTM) and the models with D-dimer alone or sTM alone are 0.007 and 0.001, respectively.

When we replicated the regression models using the propensity score sample considering only venous thrombotic events, we yielded identical outcomes (S5 Figure in S1 File). We did not analyze arterial events because there were few cases.

## Discussion

This study aimed to identify new plasma biomarkers for thrombotic events in hospitalized COVID-19 patients, with greater potential to refine risk assessment compared to D-dimer alone, to enable personalized prophylaxis for people at higher risk. Notably, sTM levels emerged as an independent marker that can improve the ability of D-dimer to identify patients at higher risk of venous and arterial events. In particular, the inclusion of sTM reduced the false positive rate associated with D-dimer alone. Angiopoietin-2 concentration correlated with sTM concentration but showed no association with thrombotic events or mortality, nor dit it add to the risk stratification for such events.

To our knowledge, this is the first study to explore the value of sTM for identifying COVID-19 patients with high thrombotic risk. Our data suggest sTM is also a prognostic predictor, as high levels were associated with mortality. Several mechanisms could explain these links. In healthy individuals, sTM is released during physiological cleavage and shedding of membrane-bound thrombomodulin, but in very low amounts [27]. In contrast, elevated sTM levels have been accepted as a reliable marker of endothelial damage [28]. With SARS-CoV-2 infection, dysfunction of the vascular endothelium and the subsequent release of TM fragments into the plasma [29] may play an important role in the pathogenesis of COVID-19 vasculopathy [30] and in the poorer outcomes observed in COVID-19 patients with higher sTM levels [31].

sTM elevation reflects the denudation of endothelial cells membranes of TM and the consequent reduction in its anticoagulant function [22]. This disruption in TM activity may contribute to the heightened thrombotic risk observed in COVID-19 patients, as the protective anticoagulant properties of TM are compromised. The sTM fragments can also inhibit fibrinolysis through the activation of thrombin-activatable fibrinolysis inhibitor [32]. In line with this hypothesis, in vitro experiments have found a lower amount of TM bound to the endothelium in cell cultures exposed to the SARS-CoV-2 spike protein [33]. The underlying mechanism is thought to involve membrane disruption and the consequent release of TM, although a downregulation mechanism of the TM-coding gene has also been proposed [34].

The rate of thrombotic events detected in our study was less than 3%, which is lower than the rates reported in other series of hospitalized COVID-19 patients [35]. In one large cohort study with a population of similar age and comorbidity [36], more than 15% of patients had arterial thrombotic events, and around 10% had venous events. This discrepancy may be attributable to our protocolized use of thrombosis prophylaxis and early detection of high risk patients, which included serial D-dimer measurements [25]. Additionally, our use of targeted, rather than universal, four-extremity duplex ultrasound may have missed asymptomatic DVT, leading to a lower observed incidence. Nevertheless, D-dimer, primarily a robust marker of coagulopathy and systemic thrombosis, is not an optimal indicator for thrombotic events. This is especially true for arterial events [20] and in scenarios with intermediate or high risk of thrombotic events, such as hospitalized COVID-19 [37], where elevated levels are common and can be due to various inflammatory processes. Furthermore, recent studies have increasingly identified endothelial dysfunction, rather than other mechanisms, as a central factor driving the pathogenesis of COVID-19 [38]. PAT events in our study predominantly affected peripheral rather than central pulmonary arteries. This pattern is commonly observed in *in situ* PAT [39], as opposed to pulmonary embolism, and is generally associated with vascular

damage in COVID-19 [40]. It may also explain the observed limitations in D-dimer and support the higher predictive ability of D-dimer combined with sTM.

There is evidence that Ang2 inhibits thrombomodulin-mediated anticoagulation and is associated with thrombotic events, including shortened bleeding time in murine models [23]. In contrast, although we found a weak but significant correlation between Ang2 levels and sTM, we found no clear association between Ang2 and thrombotic events. This may be because Ang2 has a more subtle or indirect influence on coagulation, possibly through endothelial dysfunction, without leading to clinically significant thrombus formation in all cases. Additionally, study design, sample size, and variability in thrombotic events among patients may have impacted our results.

Our findings should be considered in light of the study limitations. We conducted our investigation in a single center and identified relatively few events, which compromises the generalizability and statistical power of the results. Additionally, in some deceased individuals who died to respiratory failure progression, thrombotic phenomena not confirmed by imaging techniques may have contributed to death, leading to potential misclassification. Furthermore, biomarker measurements were taken at different time-points for thrombotic cases and controls, reflecting a real-world approach that captures data near the event but deviates from strictly predictive or diagnostic designs. This may introduce temporal bias, as elevated levels in cases could partly reflect the thrombotic process rather than pre-event risk. Finally, the low number of arterial events precluded subgroup analyses, limiting the conclusions that can be drawn from these specific outcomes. The strengths of our study include the novel findings and the representativeness of emerging SARS-CoV-2 variants throughout the entire course of the pandemic, notably our population included a high proportion of patients with the Omicron variant. All procoagulant biomarker measurements were conducted in the same laboratory using consistent techniques throughout the study period. Another notable aspect of the study is its rigorous control of biases through propensity score matching, which contributes to a more balanced analysis of the data.

In conclusion, the integration of sTM into risk stratification models may improve early diagnosis of thrombotic complications in people hospitalized with COVID-19. Our study underscores the limitations of D-dimer as a standalone marker of thrombotic events in this population, and the potential of sTM to enhance its risk-stratification capacity. Future research should focus on validating these findings in larger cohorts and exploring the mechanistic underpinnings of sTM's value in risk assessment.

## Supporting information

**S1 File. This file contains the supplementary figures and tables.**
(DOCX)

## Acknowledgements

We acknowledge Carolina Din Lin for data curation, Julia Turner for her assistance with the English language revision of this manuscript, and all study participants who made this research possible.

## Author contributions

**Conceptualization:** Sergio Padilla, María Andreo, Félix Gutiérrez, Mar Masiá.

**Data curation:** Marta Fernández-González.

**Formal analysis:** Christian Ledesma.

**Funding acquisition:** Félix Gutiérrez, Mar Masiá.

**Investigation:** Pascual Marco, Ana Marco-Rico.

**Methodology:** Sergio Padilla, María Andreo.

**Resources:** Félix Gutiérrez, Mar Masiá.

**Software:** Christian Ledesma.

**Supervision:** Félix Gutiérrez, Mar Masiá.

**Writing – original draft:** Sergio Padilla, María Andreo.

**Writing – review & editing:** Pascual Marco, Ana Marco-Rico, Christian Ledesma, Marta Fernández-González, Javier García-Abellán, Paula Mascarell, Ángela Botella, Félix Gutiérrez, Mar Masiá.

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
