## [Decision Letter · Decision Letter 0]

1 Dec 2024

PONE-D-24-44281Enhanced prediction of thrombotic events in hospitalized COVID-19 patients with soluble thrombomodulin.PLOS ONE

Dear Dr. Padilla,

Thank you for submitting your manuscript to PLOS ONE. After careful consideration, we feel that it has merit but does not fully meet PLOS ONE’s publication criteria as it currently stands. Therefore, we invite you to submit a revised version of the manuscript that addresses the points raised during the review process.

We look forward to receiving your revised manuscript.

Kind regards,

Hideto Sano

Academic Editor

PLOS ONE

Journal Requirements:

2. Thank you for stating the following financial disclosure: This work was supported by Spanish National Plan for Scientific and Technical Research and Innovation, European Regional Development Fund (ERDF) and Instituto de Salud Carlos III (RD16/0025/0038, PI16/01740, PI18/01861, CM19/00160, CM20/00066, COV20/00005, CM21/00186, CM22/00026 and PI22/01949); Consorcio Centro de Investigación Biomédica en Red (CIBER), Instituto de Salud Carlos III, Ministerio de Ciencia e Innovación and European Union – NextGenerationEU (CB21/13/00011); ILISABIO programme, UMH-FISABIO, Generalitat Valenciana (A-32 2020); Consellería de Innovación, Universidades, Ciencia y Sociedad Digital, Generalitat Valenciana (AICO/2021/205); and funded by Conselleria de Sanitat Universal i Salut Pública (Generalitat Valenciana, Spain) and the EU Operational Program of the European Regional Development Fund (ERDF) for the Valencian Community 2014–2020, within the framework of the REACT-EU program, as the Union’s response to the COVID-19 pandemic (FEDER-COVID-23).  

Reviewers' comments:

Reviewer's Responses to Questions

**Comments to the Author**

1. Is the manuscript technically sound, and do the data support the conclusions?

Reviewer #1: Partly

Reviewer #2: No

2. Has the statistical analysis been performed appropriately and rigorously? 

Reviewer #1: Yes

Reviewer #2: No

3. Have the authors made all data underlying the findings in their manuscript fully available?

Reviewer #1: Yes

Reviewer #2: No

4. Is the manuscript presented in an intelligible fashion and written in standard English?

Reviewer #1: Yes

Reviewer #2: Yes

5. Review Comments to the Author

Reviewer #1: This is a nested case control study of patients with COVID-19 and examined the utility of sTM and ang2 as predictive markers for TE.

It is well executed and well written study, however, there are some major shortcomings that directly impact the scope of this work.

- In the introduction, authors have mentioned the limitations of D dimer and that it is non specific for thrombosis. I would argue the same is true for soluble thrombomodulin. It is a marker of endothelial injury, and as a result can be elevated at the time of thrombosis, multi-organ injury, respiratory failure, etc.

- Authors mentioned about routine CTPA to rule out PAT. How about 4 extremity doppler ultrasound? If that was not done routinely with CTPA, is there a chance that an extremity DVT was missed and people were inappropriately assigned to the no thrombosis category? Please clarify.

- Authors acknowledge that D dimer is a not a great marker for arterial thrombosis in their discussion. It will be helpful to compare ROC curves for D dimer with and without sTM to truly understand if indeed sTM is a better predictor for arterial thrombosis than D dimer or not.

Reviewer #2: The authors measured serum levels of sTM, Angiopoietin-2 and D-Dimers in 73 patients developing thrombotic events (TE) within 28 days of hospital admission and 297 patients without TE.

Comments:

1. The Biomarkers were measured closest to TE in the 73 positives, but at hospital admission for the other patients. This does not work. The analysis can either aim to be predictive, then all measurements have to take place at hospital admission. Or the analysis is diagnostic, then all measurements need to take place on matched days after hospital admission, or else a time course can be shown for all patients from hospital admission to days 5 - 10 - 28 comparing patients with and without TE.

2. The Discussion emphasizes the predictive nature of the results, then values measured at hospital admission have to be shown, but removal of the 20 patients with TEs after day 28 is not justifiable.

3. This correction of timing should also include all lab values (Table 1).

4. Multivariate logistic regressions include D-Dimers, sTM, IL-6 and RT-PCR-cycle threshold. I cannot understand this selection - either include clinically meaningful variables (age, sex, comorbidity etc.) or else include all variables significant in univariable logistic regression from Table 1.

5. Figures 1 and 2: TE until day28? Overall mortality is a time dependent event, Cox regression should be used.

6. Figure 3: sTM mono is lacking – any difference to sTM+D-Dimer?

7. Figure legends are too short and do not explain what was done and what is seen. Worst example is suppl. Figure 3, I do not understand it.

8. P8 line 197: no mortality risk of PAT is shown in suppl. Table 3.

9. P10 line 223: D-Dimers did not correlate with Angoipoietin-2.

10. Figure S4 – please find an easier way of labeling the plots with rho and p-values. Always give the values for p, no stars.

11. P10 line 234: give p-value for statement.

12. P10 line 235 – mortality is a time-dependent event, what exactly is reported?

13. Discussion: all arguments relate to a predictive value of the biomarkers, but this requires measurement at hospital admission.

6. PLOS authors have the option to publish the peer review history of their article (what does this mean? ). If published, this will include your full peer review and any attached files.

**Do you want your identity to be public for this peer review?** For information about this choice, including consent withdrawal, please see our Privacy Policy .

Reviewer #1: No

Reviewer #2: No

---

## [Author Response · Author response to Decision Letter 1]

9 Jan 2025

Dear Editor,

Thank you for the comments provided by the reviewers and the Academic Editor. We have carefully addressed each point raised and have included a detailed document with our responses. This document has been uploaded to the system under the file name "Response to Reviewers."

Should you need further clarification or additional information, please do not hesitate to contact us.

Kind regards,

Sergio Padilla

---

## [Decision Letter · Decision Letter 1]

22 Jan 2025

PONE-D-24-44281R1Enhanced prediction of thrombotic events in hospitalized COVID-19 patients with soluble thrombomodulin.PLOS ONE

Dear Dr. Padilla,

Thank you for submitting your manuscript to PLOS ONE. After careful consideration, we feel that it has merit but does not fully meet PLOS ONE’s publication criteria as it currently stands. Therefore, we invite you to submit a revised version of the manuscript that addresses the points raised during the review process.

We look forward to receiving your revised manuscript.

Kind regards,

Hideto Sano

Academic Editor

PLOS ONE

**Journal Requirements:**

**Additional Editor Comments:**

Please review the reviewers’ comments and make the necessary revisions.

Reviewers' comments:

Reviewer's Responses to Questions

**Comments to the Author**

1. If the authors have adequately addressed your comments raised in a previous round of review and you feel that this manuscript is now acceptable for publication, you may indicate that here to bypass the “Comments to the Author” section, enter your conflict of interest statement in the “Confidential to Editor” section, and submit your "Accept" recommendation.

Reviewer #1: All comments have been addressed

Reviewer #2: (No Response)

2. Is the manuscript technically sound, and do the data support the conclusions?

Reviewer #1: Yes

Reviewer #2: Partly

3. Has the statistical analysis been performed appropriately and rigorously? 

Reviewer #1: Yes

Reviewer #2: No

4. Have the authors made all data underlying the findings in their manuscript fully available?

Reviewer #1: Yes

Reviewer #2: Yes

5. Is the manuscript presented in an intelligible fashion and written in standard English?

Reviewer #1: Yes

Reviewer #2: Yes

6. Review Comments to the Author

**Reviewer #1: ** (No Response)

**Reviewer #2: ** Thank you for revising the manuscript.

I still have a problem with Figure 1, see my former comment 12.

Mortality is a time dependent event. For a KW-test looking at mortality in a binary way, the time frame for mortality has to be reported (e.g. 1-year mortality, then only include patients who died before or reached the landmark, take out patients with LFU before the landmark).

Figure legends are better now, but Figure 1 shows a and b, the legend reports a-d.

7. PLOS authors have the option to publish the peer review history of their article (what does this mean? ). If published, this will include your full peer review and any attached files.

**Do you want your identity to be public for this peer review?** For information about this choice, including consent withdrawal, please see our Privacy Policy .

Reviewer #1: No

Reviewer #2: No

---

## [Author Response · Author response to Decision Letter 2]

24 Jan 2025

Reviewers' comments: 

Reviewer's Responses to Questions

REVIEWER #1: No Response

REVIEWER #2: Thank you for revising the manuscript.

I still have a problem with Figure 1, see my former comment 12.

Mortality is a time dependent event. For a KW-test looking at mortality in a binary way, the time frame for mortality has to be reported (e.g. 1-year mortality, then only include patients who died before or reached the landmark, take out patients with LFU before the landmark).

Response: Thank you for this insightful comment. We appreciate the reviewer’s attention to this important aspect. Upon re-evaluating the data, we confirm that no patients were lost to follow-up before the 28-day study censoring date. Therefore, the figure and the analyses remain unchanged. However, to address this point and ensure clarity, we have added a note below the figure stating that all patients were accounted for within the 28-day follow-up period. We trust this addition provides the necessary clarification.

Changes made in the manuscript:

Figure 1 legend: All patients were accounted for within the 28-day follow-up period, with no losses to follow-up before the censoring date.

Figure legends are better now, but Figure 1 shows a and b, the legend reports a-d.

Response: Thank you for your observation. We have revised the figure legend to clarify that each panel includes data for thromboembolic events (left) and overall mortality (right). We hope this updated legend resolves the issue.

---

## [Decision Letter · Decision Letter 2]

6 Feb 2025

Enhanced prediction of thrombotic events in hospitalized COVID-19 patients with soluble thrombomodulin.

PONE-D-24-44281R2

Dear Dr. Padilla,

We’re pleased to inform you that your manuscript has been judged scientifically suitable for publication and will be formally accepted for publication once it meets all outstanding technical requirements.

Kind regards,

Hideto Sano

Academic Editor

PLOS ONE

Additional Editor Comments (optional):

Reviewers' comments:

Reviewer's Responses to Questions

**Comments to the Author**

1. If the authors have adequately addressed your comments raised in a previous round of review and you feel that this manuscript is now acceptable for publication, you may indicate that here to bypass the “Comments to the Author” section, enter your conflict of interest statement in the “Confidential to Editor” section, and submit your "Accept" recommendation.

Reviewer #2: All comments have been addressed

2. Is the manuscript technically sound, and do the data support the conclusions?

Reviewer #2: Yes

3. Has the statistical analysis been performed appropriately and rigorously? 

Reviewer #2: Yes

4. Have the authors made all data underlying the findings in their manuscript fully available?

Reviewer #2: Yes

5. Is the manuscript presented in an intelligible fashion and written in standard English?

Reviewer #2: Yes

6. Review Comments to the Author

Reviewer #2: ok, no further comments

7. PLOS authors have the option to publish the peer review history of their article (what does this mean? ). If published, this will include your full peer review and any attached files.

**Do you want your identity to be public for this peer review?** For information about this choice, including consent withdrawal, please see our Privacy Policy .

Reviewer #2: No

---

## [Editor Report · Acceptance letter]

PONE-D-24-44281R2

PLOS ONE

Dear Dr. Padilla,

I'm pleased to inform you that your manuscript has been deemed suitable for publication in PLOS ONE. Congratulations! Your manuscript is now being handed over to our production team.

Kind regards,

on behalf of

Dr. Hideto Sano

Academic Editor

PLOS ONE